# Poor School-Related Well-Being among Adolescents with Disabilities or ADHD

**DOI:** 10.3390/ijerph19010008

**Published:** 2021-12-21

**Authors:** Sanna Tiikkaja, Ylva Tindberg

**Affiliations:** 1Centre for Clinical Research Sörmland, Uppsala University, P.O. Box 529, 631 07 Eskilstuna, Sweden; ylva.tindberg@kbh.uu.se; 2Department of Public Health and Caring Sciences, Uppsala University, 631 07 Uppsala, Sweden; 3Department of Women’s and Children’s Health, Uppsala University, 631 07 Uppsala, Sweden

**Keywords:** neurodevelopmental disorder, adolescent health, personal satisfaction, public health epidemiology, psychosocial functioning

## Abstract

Poor school-related well-being may influence adolescents’ school performance and lifestyle. Adolescents having disabilities or ADHD are in a vulnerable situation for having poor school-related well-being, compared to adolescents not having disabilities. We used cross-sectional data from a school-based survey among 15–18-year-olds (*N* = 4071) in Sörmland, Sweden, to analyse the association between poor school-related well-being and disabilities or ADHD. The analyses were carried out by logistic regression models, adjusting for background factors, school-related factors, and health-compromising behaviours. Adolescents having disabilities (*n* = 827) or ADHD (*n* = 146) reported that their disability had a negative influence on school. Compared to peers without disability, those having disabilities had an increased chance (OR = 1.40 95% CI: 1.17–1.68) of poor school-related well-being. The corresponding OR was doubled for adolescents reporting ADHD (2.23 95% CI: 1.56–3.18). For the ADHD group, the adjOR for poor school-related well-being remained significant (1.67 95% CI: 1.13–2.50) after adjustments for school-related factors and health-compromising behaviours, but not for the disability group. In conclusion, adolescents having ADHD are a particularly vulnerable group at school, having a greater risk of poor school-related well-being. Schools should actively work to achieve school satisfaction for adolescents having disabilities, to ensure that all students have similar opportunities for favourable development, health and achievement of their academic goals.

## 1. Introduction

School-related well-being is an important indicator of children’s and adolescents’ overall health. The United Nations Sustainable Development Goals (SDG) on Education focus attention on “inclusive and equitable quality education and promotion of lifelong learning opportunities for all” [1]. One of the target goals, in particular, highlights “equal educational access to persons with disabilities”. There is an ongoing discussion of how to achieve these goals in practice [2,3]. In a similar manner, the SDG on Health emphasises *“*healthy lives and promotion of well-being for all at all ages*”*. Education is seen as a means to achieving good health [4] and a tool for reducing inequalities [1]. These goals are in line with the Swedish public health politics, stating a requirement for good and equal health for all [4]. The Swedish welfare system entitles all adolescents to study at no cost, and Swedish schools shall enable all students to reach their academic goals, based on their own prerequisites [5]. Further, all students in Sweden have the right to feel safe, to make themselves heard and to feel well at school [6].

The school is the adolescents’ workplace, where they spend a significant amount of time, making school an important factor influencing adolescents’ cognitive, social and emotional development [7,8]. Feelings of having supportive teachers, being treated fairly, and feeling safe at school are related to the adolescents’ experience of the school environment [9], quality of life at school [10] and school performance [8].

A positive school experience is considered a protective factor against health-compromising behaviours [7], such as using alcohol and tobacco [11,12], and cannabis [12,13], positively influencing mental and physical health [7]. Having friends at school is an important factor in the school setting, with a potential to influence health-compromising behaviours [14] as well as academic functioning [15].

More than 20 per cent of Swedish school children have a chronic condition, such as a chronic disease, physical disability and/or neuropsychiatric functioning impairment [16]. About 10 per cent of children are estimated to have a neuropsychiatric functioning impairment such as an autism spectrum disorder (ASD) (1–3%), attention deficit/hyperactivity disorder (ADHD) (5–11%), or Tics (1%) [17,18]. Furthermore, 14 per cent report impaired hearing (7%), vision (7%) or motor function (3%) [17]. Another 8–15 per cent present with reading and/or comprehension deficit [19]. Many of these disabilities are coexisting [18].

Adolescents with disabilities often experience stigmatisation, discrimination and inequalities, such as disrespect, because of their disability [20]. Having a disability may negatively influence several aspects of the adolescents’ life, for example, relations with friends [14,21], leisure and healthy behaviour [22,23], school attendance [24], learning [5], grades [22] and school-related well-being [25,26].

The type and severity of the impairment, as well as the extent to which the impairment can be adjusted, are determinants of school outcome. Children having hearing impairment may be able to have better speech, language and functional performance if hearing devices are fitted early on [27]. ADHD is a disorder characterised by an ongoing pattern of inattention, impaired executive functioning, and/or hyperactivity and impulsivity that interferes with functioning and development [28,29]. Adolescents with ADHD may have severe difficulties at school before getting a diagnosis, which is often delayed [30] and even after receiving treatment [24]. During the life course, a disability that has not been successfully compensated for may be a barrier to completing an education, to finding suitable employment, which, in the long run, can affect income, health and opportunities for social mobility [3,31,32].

### Aim

The aim of the present school-based study is to investigate how school-related well-being is experienced by 15–18-years-old adolescents with disabilities and ADHD, respectively, compared to peers with no such impairments, while considering background factors, school-related factors, and health-compromising behaviours. There is no consistent definition of school-related well-being, and previous studies assessed the adolescents’ experiences in different ways [9,12]. In this study, school-related well-being is measured by the question “How do you like school”?

## 2. Materials and Methods

### 2.1. The Study Design and Setting

A cross-sectional study design was used with data from a tri-annual, school-based survey, Life and Health in Youth, conducted by the Department of Welfare and Public Health in collaboration with the Centre for Clinical Research at the County Council of Sörmland, Sweden. The survey targets all schools in the county of Sörmland, an area socio-economically representative of Sweden. In the beginning of 2017, the county of Sörmland had a total population of 288,097 [33]. The percentage of adolescents (15–18-year-olds) was about the same as in Sweden as a whole (6%). However, the proportion of adolescents of foreign origin was somewhat higher in the county of Sörmland compared to Sweden as a whole (23% versus 18%). The survey questionnaire was distributed to adolescents attending 9th grade (15–16-years-old) and the second year of upper secondary school (17–18-years-old), with the latter being attended by 93 per cent of all 17–18-year-olds in Sweden.

### 2.2. Data Collection

During February–March 2017, students attending 9th grade and the second year of upper secondary school in all 9 municipalities in the county of Sörmland answered the questionnaires anonymously in the classroom during school hours. School employees handed out, collected, and returned the questionnaires to the Centre for Clinical Research, Sörmland. Students who had been absent got a second opportunity to participate within two weeks. However, students and parents were informed beforehand in writing that participation was voluntary. Therefore, a completed questionnaire was regarded as the student’s informed consent. No parental approval is needed for participants above the age of 15 in Sweden [34]. The study was approved by the Regional Ethical Review Board, Stockholm (Dnr 2017/709-32).

### 2.3. Study Population and the Questionnaire

In total, 5018 adolescents in the County of Sörmland filled in the questionnaire. The response rates were 84 per cent for the 9th grade and 82 per cent for the second year of upper secondary school. The questionnaire included 78 questions for the 9th grade and 82 questions for the second year of upper secondary school regarding various aspects of the adolescents’ life, e.g., gender, lifestyle, socio-economic background, health, school, social network, family relations and child abuse. The corresponding questions were identical in the two questionnaires [35]. The Life and Health in Youth surveys have previously been used in other research projects regarding students’ health [22,36,37].

### 2.4. Measurements and Definitions

For the present study, questions on various kinds of self-reported disabilities were obtained. Based on this, three categorical groups were created. The groups were as follows:“Reference group”, including adolescents reporting none of the below conditions.“Disability group”, including adolescents answering “yes, mild” and “yes, severe” to one or more of the questions on impaired hearing, vision, and/or motor function, reading/writing impairment, Asperger’s syndrome, Autism or similar, and/or “other unspecified disabilities”.“ADHD group”, including adolescents answering “yes, mild” and “yes, severe” to ADHD, ADD, Tourette’s syndrome or similar, and did not confirm any other disability.

In total, 4570 (91%) students answered the questions on disabilities and ADHD. Of these, 222 students reported both ADHD and another disability, and were excluded. An additional 277 students were excluded due to incomplete data. In all, 4071 (1990 girls and 1935 boys) were included in the study. Of these, 426 girls and 401 boys reported disabilities, and 64 girls and 82 boys reported ADHD. In a supplementary question, adolescents having reported any kind of disabilities, including ADHD, were asked if their disability/ies had any negative influence on the following different aspects of their life: school/homework; leisure activities; relationship with friends and relationship with family, with the response alternatives “no” (“no, does not affect”), and ”yes” (“yes, affects a little”, “yes, affects quite much” and “yes, affects a lot”). In case of several reported disabilities, the question regarding influence on different aspects of life could only be answered jointly for reported disabilities. The outcome in this study was school-related well-being, measured by the question “How do you like school?” and categorised into “good” (“very good” and “good”) and “poor” (“neither good/poor”, “poor” and “very poor”).

### 2.5. Background Factors

Gender was categorised as “boy” or “girl”. Ethnicity was categorised as “Swedish” (“born in Sweden” or “having at least one Swedish parent”) or “non-Swedish” (“born outside of Sweden” or “having both parents born outside of Sweden”). A proxy for economic stress was used by the question “Are you worried about your family’s economy?”, categorised into “yes” (“yes, quite worried” or “yes, very worried”) or “no” (“not especially worried” or “not worried at all”).

### 2.6. School-Related Factors

Truancy was measured by the question “Do you usually skip going to school or to class/es?”, categorised into “yes” (“yes, a few times during the semester”, “yes, once a month”, “yes, 2–3 times a month”, “yes once a week” or “yes, several times a week”) or “no”. In order to measure passing all subjects, the question “Do you have a failing grade or do you lack grades in any subject?” was used and categorised into “yes” (“yes, in 1–2 subjects”, “yes, in 3–4 subjects” or “yes in 5 subjects or more”) or “no”. Feeling safe at school was measured by the question “Do you feel safe at the following places?” Places used were “in the classroom” and “on my way to or from school”, which were categorised into “yes” (“yes, always” or “yes, often”) or “no” (“no, seldom” or “never”). Having a friend during school breaks was measured by “Do you have someone to be with during school breaks?”, which was categorised into “yes” (“yes, often” or “yes sometimes”) or “no”.

### 2.7. Health Compromising Behaviours

Use of alcohol was measured by the question “Have you consumed alcohol during the last 12 months?”, dichotomised into “yes” (“yes, one time”, up to “yes, more than two times a week”) or “no”. For daily use of tobacco, questions on snuff use and cigarette use were combined, classified into daily use of cigarettes or snuff and not daily use (“using tobacco seldom” or “never”). Ever having used drugs was classified into “yes” (“yes, one time” or “yes, several times”) or “no”.

### 2.8. Statistical Methods

Cross tabulations were used to investigate if disabilities affected different aspects of the adolescents’ life, such as school/homework, leisure activities, relation with friends, and relation with family. Chi-Square test was used to understand the associations between background factors, school-related factors and health-compromising behaviours and disabilities and ADHD, respectively. In the next step, logistic regressions were used to examine the relation between school-related well-being and the two different groups of disabilities or ADHD. Odds ratios (OR:s) were estimated in adjusted models with 95 per cent confidence intervals (CI). *p*-values < 0.05 in two-tailed analysis were considered as statistically significant. Adjustments were made in four different steps. First, for the background factors gender, age (school grade), ethnic background and economic stress. Second, adjustments were made for school-related factors (having incomplete grades, truancy, feeling unsafe in the classroom, feeling unsafe on the way to or from school, and not having a friend during breaks at school). Third, there were adjustments for health-compromising behaviours (having consumed alcohol during the last 12 months, ever having used drugs and daily tobacco use). Fourth, adjustments were made for all confounders used in the models. All data analyses were carried out using SPSS 22.0 for Widows (SPSS Inc., Chicago, IL, USA).

## 3. Results

Three of five students reported no disability (control group). As shown in Table 1, the control group consisted of somewhat larger proportion of students in 9th grade than the second year of upper secondary school, while the proportions of boys and girls were similar. Most of the students in the control group had Swedish background, reported no economic stress, a good overall situation at school and reported healthy behaviours (Table 1). Among the 1093 excluded students, the proportion of boys (52%) was similar as for those included, albeit lower for girls (33%), while 15% of the excluded students lacked information on gender. Slightly more students were excluded from 9th grade (56%) than from the second year of upper secondary school (44%). Excluded students had the same proportion of non-Swedish background (25%) as those included, while 55 per cent stated that they were of Swedish origin; 20 per cent lacked data. Excluded students more often reported economic stress (20%), compared to those included, while 7 per cent of excluded students lacked this information.

### 3.1. Disabilities

In all, 20 per cent of the students reported one or more disabilities (disability group), and four per cent reported that they had ADHD (ADHD group). As shown in Table 1, disabilities and ADHD were more common among boys than girls. Most adolescents reporting disabilities or ADHD were of Swedish origin. Adolescents reporting disabilities or ADHD were also more likely to have experiences of economic stress than peers without disabilities. Among adolescents within the disability group, 69 per cent stated that the disability had negative impact on school/homework, a statement that was even more common among the ADHD group (87%). The corresponding negative influence on the other aspects of life were considerably lower, as follows: leisure activities (28% and 38%), relationship with friends (21% and 43%), and relationship with family (21% and 43%). Therefore, we focus on school-related well-being in the present paper. Boys, adolescents of Swedish origin and adolescents having experiences of economic stress, were overrepresented among participants reporting disabilities, especially those reporting ADHD (Table 1). No association was seen between the two school years among 15–18-year-olds and the prevalence of disabilities or ADHD. Compared to the control group with no impairments, the disability group more often reported incomplete grades, truancy behaviour, feeling unsafe in the classroom and on the way to and from school, not having a friend during breaks at school, as well as poor school-related well-being (Table 1). The ADHD group reported even higher rates of truancy and poor school-related well-being (Table 1). All health-compromising behaviours, using alcohol or tobacco and having tried drugs, were more frequent among adolescents in the disability group compared to peers with no disability. The associations were even stronger for the ADHD group.

### 3.2. Poor School-Related Well-Being

One in five students with no disabilities reported poor school-related well-being, compared to one in four of the disability group and one in three in the ADHD group (Table 1). Compared to the control group, the OR:s for reporting poor school-related well-being was 1.40 (95% CI: 1.17–1.68) for the disability group and 2.23 (95% CI: 1.56–3.18) for the ADHD group (Table 2). These OR:s for poor school-related well-being remained similar after adjustment for background factors. The OR for the disability group was 1.41 (95% CI: 1.17–1.69) and 2.23 (95% CI: 1.56–3.32) for the ADHD group (not shown in Table 2). In the second step, adjustments were made for background factors and school-related factors, resulting in non-significant OR:s for poor school-related well-being in the disability group (adjOR = 1.13, 95% CI: 0.92–1.38), and somewhat lowered but significant OR:s for the ADHD group (adjOR = 1.88, 95% CI: 1.26–2.80). In the third step, with adjustments for background factors and health-compromising behaviours, the OR:s for poor school-related well-being was significantly elevated for the disability group (adjOR = 1.36, 95% CI: 1.13–1.64) as well as the ADHD group (adjOR = 1.88, 95% CI: 1.30–2.72). In the fully adjusted model (step 4), considering background factors, school-related factors and health-compromising behaviours, the OR:s for poor school-related well-being did not reach statistical significance for the disability group, while the OR:s for poor school-related well-being remained significant for the ADHD group (adjOR = 1.67, 95% CI: 1.13–2.50).

## 4. Discussion

In the present school-based study, 15–18-year-old students reporting disabilities or ADHD were significantly more likely to report poor school performance, low school attendance, feeling unsafe at school as well as on the way to and from school, not having a friend during breaks at school, poor school-related well-being, and higher engagement in health-compromising behaviours than peers without these impairments. For adolescents reporting disabilities, not including ADHD, the increased risk for poor school-related well-being disappeared when controlling for other school-related factors but not for health-compromising behaviour. Adolescents with ADHD, on the other hand, showed a significantly increased risk for poor school-related well-being when controlling for both school-related factors and health-compromising behaviours.

When focusing on school-related well-being, we noted a strong association between adolescents with ADHD and poor school-related well-being, even when controlling for both school-related factors and health-compromising behaviours. For students with other disabilities, this association disappeared after adjustments for school-related factors, but not for health-compromising behaviours. This is, in part, in line with previous studies showing that students having disabilities, such as impairment in vision, hearing, and motor functioning as well as neuropsychiatric impairments, had significantly lower feeling of satisfaction and well-being at school than students without a disability [17,25,26]. There may be several explanations for our findings, as follows:

In the current study, having incomplete grades was significantly associated with both self-reported disabilities (37%) and ADHD (38%) compared to peers with no such impairments (20%). The prevalence of lower academic performance is likely to vary within the respective groups, depending on type and degree of impairment as well as compensating adjustments. For instance, students with hearing impairments are likely to have a hearing device [27], while students with reading or writing impairments [38] or ADHD [30] may not yet have been identified and given adequate compensating aid. Adolescents with ADHD are known to perform less well academically [39,40] and are perceived to have more externalised behaviours by teachers and parents [39], as opposed to peers without ADHD. Furthermore, adolescents with ADHD may have severe difficulties at school, even after receiving treatment [24].

Our finding of a high overrepresentation of students with incomplete grades among students with disabilities or ADHD contradicts the requirements stated by the National Agency for Education. Specifically, they state that Swedish schools are obliged to ensure that all students reach their academic goals, based on their own prerequisites [5]. The results highlight the need for measures to improve students’ school and learning environment, resulting in better and more equally distributed school-related well-being.

Our study shows that students with disabilities, especially ADHD, were significantly more likely to be truant. This finding is congruent with previous results showing that involvement in school is lower among students having ADHD [41]. It is likely that negative attitudes towards people with disabilities may play a role in their school satisfaction [20], thus negatively affecting attendance at school. It has previously been found that students’ feelings of having supportive teachers who treat them fairly are important for school satisfaction and performance [9]. However, both teachers and parents often have negative perceptions of students having ADHD, while the students’ own self-perceptions of skills and interest in reading do not differ among those with and without ADHD [39].

Feeling safe at school and having friends at school were significantly underrepresented by adolescents reporting disabilities or ADHD in the present study. Both factors are important in enabling students to have positive school experiences (school-related well-being) [8,9,24]. Having friends is also a protective factor against health-compromising behaviours [14]. For adolescents with disabilities or ADHD, both friends and feelings of safety may be a particular challenge, since they often experience stigmatisation, discrimination and inequalities, such as disrespect, because of their disability [20]. During adolescence, exploring and developing one’s own identity and norms, as well as fitting in to the group become central, which as a consequence may lead to exclusion of those who are different [42]. Having ADHD, alone or in combination with another chronic condition, may increase the negative impact on protective factors at school, leisure and home [36].

In the present study, consumption of alcohol, daily tobacco usage and ever having tried drugs were more often reported by the disability group, and in particular by the ADHD group, compared to the control group without disabilities. Experimenting with health-compromising behaviours is part of normal psycho-social development during adolescence and also part of being accepted by peers, especially if having a disability [22]. These aspects may be even more challenging if the adolescent has impaired impulse control and hyperactivity, as in the case of ADHD [36]. A positive school experience is a known protective factor against health-compromising behaviours [8,9,10], positively influencing mental and physical health [7]. Therefore, efforts that strengthen protective school factors, such as school related well-being, are important complements to health education, addressing healthy lifestyles and health-compromising behaviours. Schools should employ a range of available strategies including the delivery of life skills for health and well-being, comprehensive sexuality education, and support from a positive school ethos [43]. For adolescents, schooling that is adapted to maturity and physical and mental conditions is a means to good health [5] and a tool for reducing inequalities [1]. In Sweden, schools are obliged to have a health-promoting and risk-preventive perspective, with focus on strengthening the student’s protective factors, leading to good health and learning [44].

Even though Sweden is a forerunner of inclusive education and for having a clear agenda for rights and equity for people with disabilities, exclusion still occurs in the educational system [2]. Taneja-Johansson and Singal point out existing disability discrimination related to education, school absenteeism and educational segregation that needs to be improved. A twin-track approach should be promoted, ensuring that mainstream educational programmes are designed for all learners with simultaneous developing of targeted support to address the specific needs of children with disabilities [2]. Furthermore, children with disabilities, particularly ADHD, may benefit from school and parents working together, particularly for children having ADHD, as parents may help to regulate behaviours influencing both school-related factors and health-compromising behaviour [2,3].

### Strengths and Limitations

One limitation of this study is the self-reported data, particularly on disabilities and ADHD. The present study included 4071 students, of which 20 per cent stated that they had a disability. Nevertheless, the prevalence of the disabilities in the current study is in good agreement with previously reported data [16,17,18]. In addition, self-reported data with information on the students’ own school experiences is a feasible strategy for a large-scale school-based survey. However, there are other unmeasured factors that may influence the students’ experience of school-related well-being, such as class size and relationship with the teacher, which we could not control for in our study. Furthermore, the ana-lysis on excluded students showed a greater proportion of students experiencing economic stress (20% vs. 12%), indicating that we might underestimate the impact of economic stress on poor school-related well-being. The strengths of the study are the school-based study design and the large study size, enabling powerful analyses on school-related well-being and disabilities. Another strength is the data collection made at school, and the students anonymously answering the questionnaire including sensitive questions, such as having disabilities, economic stress and health-compromising behaviours. Finally, the population-based study design may allow for generalisation to Swedish students and, with some caution, to other similar populations.

## 5. Conclusions

School is an important setting for promoting adolescents’ health and well-being. Safeguarding young people’s rights and opportunities for a safe, nurturing and satisfying schooling is one of societies and the school’s major challenges. However, this basic precondition is not safeguarded for all students on equal terms. The present study shows a significant discrimination of students with disabilities or ADHD in the school-setting, with lower academic performance, presence and involvement, feelings of safety and having a friend. Students reporting ADHD also had a significantly higher risk of poor school-related well-being. Altogether, schools would benefit greatly from working to reduce the effects of negative factors for adolescents with disabilities or ADHD. A holistic supportive approach should involve several stakeholders, such as the student health team and school personnel, health care professionals, parents and the students themselves. This may help to identify and strengthen the possibilities for students with disabilities, especially those with the challenges of ADHD, to fulfil their academic goals, gain health and succeed in life.

## Figures and Tables

**Table 1 ijerph-19-00008-t001:** Background factors, school-related factors and health-compromising behaviours and their association to self-reported disabilities, or ADHD.

Sample Characteristics	Healthy	Disability	ADHD	*p*-Value *
	*N* = 3098 *n* = %	*N* = 827 *n* = %	*N* = 146 *n* = %	
**Background factors**				
**Gender** Boy Girl	1509 (49) 1589(51)	426 (52) 401 (48)	82 (56) 64 (44)	0.09
**Grade** 9th grade (15–16 y) Year 11 upper secondary school (17–18 y)	1660 (54) 1438 (46)	427 (52) 400 (48)	83 (57) 63 (43)	0.41
**Ethnicity** Swedish Non-Swedish	2265 (73) 833 (27)	637 (77) 190 (23)	123 (84) 23 (16)	0.01
**Economic stress** No Yes	2729 (88) 369 (11)	701 (75) 126 (15)	120 (82) 26 (18)	0.01
**School-related factors**				
**School-related well-being** Good Poor	2511 (81) 587 (19)	623 (75) 204 (25)	96 (66) 50 (34)	0.00
**Incomplete grades** No Yes	2470 (80) 628 (20)	522 (63) 305 (37)	90 (62) 56 (38)	0.00
**Truancy** No Yes	2887 (94) 211 (7)	746 (90) 81 (10)	119 (82) 27 (19)	0.00
**Feeling unsafe in classroom** No Yes	2559 (83) 539 (17)	746 (75) 196 (24)	110 (75) 36 (25)	0.00
**Feeling unsafe to and from school** No Yes	2391 (77) 707 (23)	582 (70) 245 (30)	100 (69) 46 (32)	0.00
**Not having a friend during breaks at school** No Yes	3049 (98) 49 (2)	793 (96) 34 (4)	142 (97) 4 (3)	0.00
**Health-compromising behaviours**				
**Alcohol consumption** No Yes	1510 (49) 1588 (51)	351 (42) 476 (58)	38 (26) 108 (74)	0.00
**Daily tobacco use** No Yes	2891 (93) 207 (7)	733 (89) 94 (11)	109 (75) 37 (25)	0.00
**Ever used drugs** No Yes	2879 (93) 219 (7)	746 (90) 81 (10)	116 (80) 30 (20)	0.00

* Statistically significant values as follows: *p* < 0.05.

**Table 2 ijerph-19-00008-t002:** The association between poor school-related well-being and self-reported disabilities or ADHD.

Disability	OR * (CI) Step 1	Adj ** OR (CI) Step 2	Adj *** OR (CI) Step 3	Adj **** OR (CI) Step 4
Disability group Yes No				
	1.40 (1.17–1.68)	1.13 (0.92–1.38)	1.36 (1.13–1.64)	1.11 (0.90–1.36)
	Ref.	Ref.	Ref.	Ref.
ADHD group Yes No				
	2.23 (1.56–3.18)	1.88 (1.26–2.80)	1.88 (1.30–2.72)	1.67 (1.13–2.50)
	Ref.	Ref.	Ref.	Ref.

* adjusted for adolescents’ background factors ** adjusted for adolescents’ background factors + school-related factors. *** adjusted for adolescents’ background factors + health-compromising behaviours. **** adjusted for adolescents’ background factors + school-related factors + health-compromising behaviours.

## Data Availability

The dataset generated and analysed in the current study is available to the authors. But are not publicly available due to ethical guidelines.

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
