# Peer review of "Poor School-Related Well-Being among Adolescents with Disabilities or ADHD"

_ijerph, 2021, doi:10.3390/ijerph19010008_

Round 1
Reviewer 1 Report
The present study is relevant because it deals with a topic of interest for the school community such as school well-being among adolescents with disabilities or ADHD and how their vulnerable situation may influence their poor school performance and lifestyle. In addition, this study adds interest by addressing this issue by analysing research results in comparison with other adolescents who do not have disabilities.
To conduct this research, the authors draw on an impressive sample of more than 4000 young people in Sweden aged 15-18 years. The results showed that having a disability has a negative influence on school and, in addition, in the case of adolescents with ADHD it also influences behaviours that compromise their own health. These findings confirm the need for schools to ensure equal opportunities for all students.
Furthermore, the study design is cross-sectional and interesting in that it takes into account a quantitative approach in which a large number of young people are surveyed on public well-being and health well-being. I consider that the representativeness of the sample is justified by the selection of this sample size. The selection of independent variables is correct and allows us to generate interesting statistical crosstabs. Likewise, the statistical analyses used are correct and refer to the search for association of the antecedent factors, those related to school and, finally, those that compromise health.
On the other hand, the theoretical framework of the study presents an interesting review of primary and secondary scientific sources. However, although the paper succeeds in addressing a large number of references and studies on this topic, it is recommended that the authors conduct a more exhaustive search on current related research to improve the discussion of the manuscript. In the last two years, manuscripts on this topic have increased in line with the Sustainable Development Goals and the 2030 Agenda. Therefore, it will be possible to find scientific evidence to enhance the theoretical framework and discussion.
Without further ado, congratulations on the work done.
Reviewer 2 Report
ABSTRACT
It is well structured and written in a concise and easy-to-read form, highlighting the main points of the article. I only suggest that where the authors put “…increased OR (1.40 95% CI: 1.17–1.68)....”, an increased chance be placed.
INTRODUCTION
It presents the study problem in a concise and well-structured manner, placing the significance of the study on the basis of relevant literature and defining the objectives of the study.
METHODS
They were described in a detailed and clear way.
RESULTS
The results are well organized and well presented in tables. However, there are some inconsistencies that need to be corrected.
I suggest that the title of Table 2 be revised because it is not necessary to put the adjustment variables in the title.
REFERENCES
References are updated. I suggest reviewing the journal's rules and checking all references.
Reviewer 3 Report
Dear authors, congratulations for this important study and manuscript.
I have some improvement suggestions:
1) Review text edition (there are differents types of text formating maybe related with copy-paste, for e.g. line 59;
2) Many references number in the text are with a too big letter size, comparing with the rest of the text.
3) In your conclusions, I believe all your results deserve some recomendations considering the protective issues you have identified, that can be improved and developed by health and education authorities.
These are litle improvements that I believe that can make your manuscript perfect. :) Just a litle more effort and you can do it!
